# Functional Study of the Role of the *Methyl Farnesoate Epoxidase* Gene in the Ovarian Development of *Macrobrachium nipponense*

**DOI:** 10.3390/ijms25137318

**Published:** 2024-07-03

**Authors:** Mengying Zhang, Sufei Jiang, Wenyi Zhang, Yiwei Xiong, Shubo Jin, Jisheng Wang, Hui Qiao, Hongtuo Fu

**Affiliations:** 1Wuxi Fisheries College, Nanjing Agricultural University, Wuxi 214081, China; zmy20000310@163.com (M.Z.); jiangsf@ffrc.cn (S.J.); wangjs0808@163.com (J.W.); 2Key Laboratory of Freshwater Fisheries and Germplasm Resources Utilization, Ministry of Agriculture and Rural Affairs, Freshwater Fisheries Research Center, Chinese Academy of Fishery Sciences, Wuxi 214081, China; zhangwy@ffrc.cn (W.Z.); xiongyw@ffrc.cn (Y.X.); jinsb@ffrc.cn (S.J.)

**Keywords:** *Macrobrachium nipponense*, *Methyl farnesoate epoxidase*, ovarian development, RNAi

## Abstract

*Methyl farnesoate epoxidase* (*MFE*) is a gene encoding an enzyme related to the last step of juvenile hormone biosynthesis. *Mn-MFE* cDNA has a total length of 1695 bp and an open reading frame (ORF) length of 1482 bp, encoding 493 amino acids. Sequence analysis showed that its amino acid sequence has a PPGP hinge, an FGCG structural domain, and other structural domains specific to the P450 family of enzymes. *Mn-MFE* was most highly expressed in the hepatopancreas, followed by the ovary and gill, weakly expressed in heart and muscle tissue, and barely expressed in the eyestalk and cranial ganglion. *Mn-MFE* expression remained stable during the larval period, during which it mainly played a critical role in gonadal differentiation. Expression in the ovary was positively correlated and expression in the hepatopancreas was negatively correlated with ovarian development. In situ hybridization (ISH) showed that the signal was expressed in the oocyte, nucleus, cell membrane and follicular cells, and the intensity of expression was strongest at stage O-IV. The knockdown of *Mn-MFE* resulted in a significantly lower gonadosomatic index and percentage of ovaries past stage O-III compared to the control group. However, no differences were found in the cumulative frequency of molting between the experimental and control groups. Moreover, the analysis of ovarian tissue sections at the end of the experiment showed differences between groups in development speed but not in subcellular structure. These results demonstrate that *Mn-MFE* promotes the ovarian development of *Macrobrachium nipponense* adults but has no effect on molting.

## 1. Introduction

*Macrobrachium nipponense* (oriental river prawn) lives in freshwater, including rivers, lakes, ponds, and marshes. Originally from the Indo-Pacific, it is now distributed around the world, and it is an economically important species in China [1,2,3]. *M. nipponense* reaches sexual maturity rapidly. During the breeding season (April–October), hatching to ovarian development and maturity occurs within about 45 days [4]. This phenomenon leads to the coexistence of multiple generations and decreased stress resistance and survival rates, which seriously affect the economic benefits of breeding. The problem of rapid sexual maturation during production has plagued the sustainable development of the breeding industry for many years [5]. In order to analyze the phenomenon of “rapid sexual maturity”, we screened the signaling pathway of “insect hormone biosynthesis” by means of KEGG enrichment from the comparative transcriptome of the hepatopancreas at different periods of the ovary of *M. nipponense* in a previous study [6]. This pathway plays a very important role in the whole ovarian development and the *methyl farnesoate epoxidase* (*MFE*) gene is significantly enriched in this pathway.

Methyl farnesoate (MF) is synthesized from the mandibular organs of crustaceans [7]. The chemical structure of MF is almost identical to that of insect juvenile hormone (JH) III except that it lacks an epoxy group [8]. In crustaceans, MF exhibits similar characteristics to JH III in insects, and it is thought to be involved in the regulation of anti-metamorphosis and ovarian development [9,10,11,12,13,14,15,16]. MF has multiple functions in crustaceans. For example, it can affect the sex ratio of *Daphnia* offspring by programing embryos to develop into males [17]. Fu et al. reported that MF induced lipid accumulation in the hepatopancreas of *Scylla paramamosain* to prepare for ovarian development [18], and Raghavan et al. found that MF is effective in inducing molting of the *Freshwater crab* [19]. Muhd-Farouk et al. reported that orally administered MF could boost the production of the mature ovary in *Scylla olivacea* [20].

The *MFE* gene encodes the enzyme related to the last step of JH biosynthesis [21,22]. The MFE enzyme can epoxidize farnesoic acid (FA) or MF into JH III acid or JH III. The *MFE* gene has been found in desert locust *(Schistocerca gregaria)* [23], German cockroach *(Blattella germanica)* [24], Pacific beetle cockroach *(Diploptera punctata)* [25], red flour beetle *(Tribolium castaneum)* [26], and other species. In previous reports about *Blattella germanica* [24] and *Diploptera puncata* [25], researchers have referred to *MFE* as *CYP15A1*. *CYP15A1* encodes a cytochrome P450 enzyme that also epoxidizes MF into JH [26]. *CYP15A1* was isolated and identified for the first time in *D. punctata*, in which it is specifically expressed in the corpus allatum (CA) [25]. In *T. castaneum*, *CYP15A1* is expressed at higher levels in the embryonic stage and in the middle of the final larval instar stage [26]. This result confirmed that *CYP15A1* is involved in the biosynthesis of JH. Functional verification revealed that the knockdown of the gene did not cause precocious pupation, indicating that MF and JH may have similar activities [26]. In the blister beetle (*Epicauta chinensis*), the expression of *MFE* in the head and reproductive system is high, suggesting that *MFE* transcript levels may be more relevant to the synthesis of JH-III than cantharidin [27]. The relative expression level of *MFE* was the highest in the first instar larva, whereas the expression level was much lower from the second to the fifth instar larvae. Compared with the control group, the content of cantharidin significantly decreased after knocking down *MFE*, which indicated that *MFE* was involved in the biosynthesis of cantharidin [27]. Similar sequences related to crustaceans can be searched on the National Center for Biotechnology Information (NCBI) database, but in-depth functional studies are lacking. 

In this study, we cloned the *MFE* gene screened from the hepatopancreas transcriptome and analyzed the sequences for their structure and evolution. We assessed the expression pattern of *Mn-MFE* mRNA in different tissues, developmental stages, and adult ovarian stages using quantitative PCR (qPCR). We used in situ hybridization (ISH) to detect the subcellular localization of *Mn-MFE* during the following five periods of ovarian development: O-I, undeveloped stage; O-II, developing stage; O-III, nearly ripe stage; O-IV, ripe stage; and O-V, spent stage. We also used RNA interference (RNAi) technology to knock down *Mn-MFE* and then evaluated its function in ovarian development by analyzing interference efficiency, gonadosomatic index (GSI), ovarian development period, tissue sections, and cumulative molting frequency. The rapid ovarian development of female *M. nipponense* is an important characteristic of its rapid sexual maturity, which seriously restricts the sustainable development of the culture of this species. Our results provide important data relevant to the analysis of the molecular mechanism of ovarian development in crustaceans, and they provide a reference for solving the problem of rapid sexual maturation during production.

## 2. Results

### 2.1. Sequence Analysis

The open reading frame (ORF) of *Mn-MFE* was validated using PCR sequencing (Figure 1). The full-length *Mn-MFE* cDNA sequence was 1695 bp, and the ORF length was 1482 bp (Genbank accession No.: PP582172). The ORF encoded 493 amino acids, and the lengths of the 5′ non-coding region and 3′ non-coding region were 44 bp and 169 bp, respectively. The molecular weight (MW) of the Mn-MFE protein was 55.81 kDa and the theoretical isoelectric point (PI) was 6.12. Signal peptide prediction results showed that there was a signal peptide of type SP(Sec/SPI) at the position of amino acids 23–24 of Mn-MFE, and the first 23 amino acids were MLIVVALVLLATLLYFNVRKPEG. In addition, no transmembrane domain was found in the amino acid sequence of Mn-MFE. The sequence analysis showed that the amino acid sequence contained the functional domains of cytochrome P450 at loci 21–38, 52–71, 106–148, 166–184, 280–297, 300–326, 343–361, 384–408, and 438–461. A comparative analysis of the Mn-MFE amino acid sequences with those of other species of MFE revealed the presence of a signature heme-binding region (FGCG).

### 2.2. Sequence Comparison and Phylogenetic Analysis

We used DNAMAN 6.0 software to compare the amino acid sequence of the Mn-MFE protein with that of other crustacean species and insect species (Figure 2), which have a typical PPGP hinge homologous to microsomal cytochrome P450 enzyme as well as a signature heme-binding region (FGCG). They also have the following conserved structural domains specific to the P450 family: Helix-C (WxxxR), Helix-I (AGxxT), Helix-K (ExxR), and PERF (PxxFxPxRF). The amino acid similarity between Mn-MFE and the MFEs of Chinese mitten crabs (*Eriocheir sinensis*), American lobster (*Homarus americanus*), snow crab (*Chionoecetes opilio*), red swamp crayfish (*Procambarus clarkii*), and Australian freshwater crayfish (*Cherax quadricarinatus*) was 61.74%, 58.79%, 58.50%, 58.38%, and 57.09%, respectively. The amino acid similarity with the parasitic wasp (*Diachasma alloeum)*, yellow fever mosquito (*Aedes aegypti*), *S. gregaria*, and *B. germanica* was 38.07%, 37.28%, 35.93%, and 32.82%, respectively. The similarity between Mn-MFE and those of the above listed crustaceans is >50%, indicating that MFE is highly conserved in crustaceans. The phylogenetic evolutionary tree of Mn-MFE amino acid sequences shows that all sequences were divided into two major branches of crustaceans and insects (Figure 3). Prawns, lobsters, crabs, and *M. nipponense* clustered together into one large branch of Crustacea, whereas insects clustered into a different large branch.

### 2.3. Expression Analysis of Mn-MFE

According to the results shown in Figure 4, the expression level of *Mn-MFE* was highest in the hepatopancreas (*p* < 0.05), followed by the ovary and gill. It was weakly expressed in heart and muscle tissue and was hardly expressed in the eyestalk and cerebral ganglion. 

qPCR results showed that *Mn-MFE* mRNA was expressed and transcribed in both embryonic and larval development stages of female *M. nipponense* (Figure 4). The different developmental stages we monitored were divided into three stages. The first stage was the embryonic development period (from CS to ZS), during which the expression of *Mn-MFE* mRNA was stable (*p* > 0.05). The second stage was postembryonic development (from L1 to L15), during which the expression gradually increased, but the difference was not significant (*p* > 0.05). The third stage encompassed metamorphosis (from PL1 to PL25), and *Mn-MFE* expression significantly increased and peaked on the 20th day after metamorphosis (PL20) (*p* < 0.05). After reaching the maximum value, the expression level significantly decreased (*p* < 0.05).

We detected *Mn-MFE* mRNA expression at five periods of ovarian and hepatopancreas development (Figure 4). However, with the gradual development and maturation of the ovary, the expression of *Mn-MFE* mRNA in the hepatopancreas significantly decreased. At the most mature stage of the ovary (O-IV), the expression of *Mn-MFE* mRNA in the hepatopancreas decreased to the lowest level (*p* < 0.05). This was followed by a gradual increase (*p* < 0.05) of expression from ovarian emptying onwards, and its expression pattern was negatively correlated with ovarian development. In contrast, the expression pattern of *Mn-MFE* in the ovary was significantly positively correlated with ovarian development. The expression significantly increased (*p* < 0.05) as the ovary matured, reaching a maximum at the most mature stage (O-IV). After ovarian emptying, the expression dramatically decreased (*p* < 0.05).

### 2.4. Localization of Mn-MFE in Different Stages of Ovarian Development

The ISH results showed that *the Mn-MFE* signals were generally weak in all of the ovarian developmental stages (Figure 5). In all stages, *Mn-MFE* probe signals were expressed in oocytes, the nucleus, the cell membrane, and follicular cells. As ovary development progressed, the intensity of *Mn-MFE* signaling increased, and it was relatively strong at stage O-IV. This result is consistent with the qPCR results for different developmental periods of the ovary.

### 2.5. Functions of Mn-MFE in Regulating Ovarian Development

We used RNAi to further investigate the function of *Mn-MFE* in the process of ovarian development. The results of short-term interference experiments are shown in Figure 6. On the 1st day after injection, the expression of *Mn-MFE* mRNA showed almost no difference between the two groups (*p* > 0.05). On the 4th and 7th day after injection, the expression of *Mn-MFE* in the experimental group was significantly lower than that of the control group (*p* < 0.05), with an interference efficiency of 85.30% and 91.29%, respectively. These results clearly showed the interference effect of dsRNA.

The GSI gradually increased in both groups as the ovaries developed (Figure 7A). On the 1st day after injection, GSI was 1.51 ± 0.03% in the control group and 1.65 ± 0.16% in the experimental group, with no significant difference between the two groups (*p* > 0.05). On the 7th day after injection, the difference between the control group (2.77 ± 0.83%) and the experimental group (2.33 ± 0.38%) was not statistically significant either (*p* > 0.05). However, on the 14th day after injection, the GSI of the control group (4.34 ± 0.48%) was significantly higher than that of the experimental group (3.31 ± 0.30%) (*p* < 0.05). On the 21st day after injection, the GSI of the control group (4.65 ± 0.15%) was also significantly higher than that of the experimental group (3.54 ± 0.13) (*p* < 0.05).

Figure 7B shows the proportion of prawns with ovarian development past stage O-III. At the beginning of the experiment, ovarian development was in stage O-I, but as the experiment progressed, the proportion of prawns that had passed the O-III stage showed a gradual increasing trend. On the 7th day after injection, there was no significant difference between the control group (5.26 ± 0.35%) and the experimental group (4.57 ± 0.31%) (*p* > 0.05). On the 14th day after injection, however, 7.02 ± 0.15% of the control group and 5.13 ± 0.12% of the experimental group passed the O-III stage, and the difference between the two groups was significant (*p* < 0.05). The proportion of specimens past stage O-III in the experimental group was similar between day 7 and day 14. On the 21st day after injection, the percentage of the control group past stage O-III (15.38 ± 0.33%) was significantly higher compared to that of the experimental group (8.77 ± 0.23%) (*p* < 0.05).

At the end of the experiment, most of the ovaries of the prawns were in the O-II stage. We randomly sampled O-II ovaries from both groups for paraffin section observation and analysis (Figure 7C). No abnormal changes in the subcellular structure of ovarian tissue were observed in the experimental group compared to the control group. We also monitored molting and found no difference between the groups in the cumulative molting frequency. These results indicated that RNAi did not affect the molting of *M. nipponense* (Figure 7D).

## 3. Discussion

*Methyl farnesoate epoxidase* (*MFE*), which epoxidizes methyl farnesoate (MF) to juvenile hormone (JH) [28,29], is a gene that is significantly enriched in the insect hormone biosynthesis signaling pathway. To date, the function of *MFE* has not been extensively studied, and there are only a few reports of its function in insects such as *B. germanica* [24] and *D. punctata* [25]. Similar sequences of *MFE* from crustaceans have been deposited in the NCBI database, but studies of the function of this gene are lacking.

The amino acid sequence of Mn-MFE contains a signal peptide with transport. The absence of a transmembrane structural domain indicates that it is not a membrane protein localized in biological membranes. Helix-C (WxxxR), Helix-I (AGxxT), Helix-K (ExxR), and PERF (PxxFxPxRF) are considered to be the four conserved domains unique to the P450 family [30,31,32]. Additionally, PPGP is thought to be a typical hinge homologous to microsomal cytochrome P450 enzyme [33,34], and FGCG is a signature heme-binding region [35]. A comparison of the MFE amino acid sequences derived from the NCBI database revealed that the PPGP hinge and the specific conserved domains of the P450 family are quite conserved among all species. However, the structural domain of FGCG in *M. nipponense* has nine more amino acids in the middle position compared to the structural domains of FGCG in other species. This suggests that the structural domain may have been altered in *M. nipponense*, resulting in a difference in function. The amino acid sequence comparison revealed that the similarity of Mn-MFE to the amino acids of all other crustacean MFEs exceeded 50%, whereas most of the amino acid similarities with insect MFEs were around 30%. This result suggests that the MFE protein is evolutionarily conserved in crustaceans.

Most insect MFE*s* are specifically expressed in the CA, such as the *CYP15* gene in the *D. punctata*, silk moth (*Bombyx mori*), *B. germanica*, and *S. gregaria* [24,25,26,36,37,38]. Expression in tissues other than CA has also been reported. Examples include the male appendage glands of the cecropia moth (*Hyalophora cecropia*) and *A. aegypti* [39,40]; the imaginal discs of the moth (*Manduca sexta*) [41]; the head and ovary-less reproductive system of *E. chinensis* [27]; and the testes, male appendage glands, and ovary of longhorn beetles (*Apriona germari*) [42]. This suggests that there may also be significant differences in the function of *MFE* genes in insects. In crustaceans, such as female gazami crabs (*Portunus trituberculatus*), *CYP15* mRNA is highly expressed in the hepatopancreas and ovary [13]. Our qPCR results showed that *Mn-MFE* mRNA expression was highest in the hepatopancreas, followed by the ovary. We hypothesize that there may be significant functional differences in *MFE* between crustaceans and insects and that the more distant the species are from each other, the more obvious the functional differences are.

The relative expression of *E. chinensis MFE* (*Ec-MFE*) was highest in first instar larvae, and the value was 2.5 times higher than that of eggs. However, the expression level of *Ec-MFE* in second–fifth instar larvae was <4% of that of first instar larvae [27]. The expression of *T. castaneum CYP15* mRNA remained low in the larval stage, began to increase on day 2 of the final larval stage, and peaked on day 3. It rose before molting into the adult stage occurred and then decreased to low levels in adulthood [26]. Cardoso-Júnior et al. reported that throughout the third instar stage of the stingless bee (*Melipona scutellaris*), *MFE* expression levels were low and only increased slightly when larvae entered the metamorphosis stage [43]. The expression of *Mn-MFE* mRNA at different developmental periods in our study significantly differed from that of *E. chinensis* and was more similar to that of *T. castaneum* and *M. scutellaris*. The expression level of *MFE* was moderate and stable at the embryonic development stage and after hatching stage, and the expression level increased significantly on the 20th day after metamorphosis. At this stage, the gonads of *M. nipponense* began to differentiate, suggesting that *MFE* might be involved in the differentiation of gonads. *Mn-MFE* expression remained stable during the larval period and may play an important role mainly after metamorphosis, especially during gonadal differentiation.

The expression of *Mn-MFE* in the hepatopancreas and ovary at five stages was further studied. The results showed that the expression of *Mn-MFE* in the ovary was significantly positively correlated with ovarian development, while it was not the same in the hepatopancreas. There are no reports of *MFE* being involved in ovarian development in insects. However, MF has been reported to be multi-functional in crustaceans. In *Daphnia*, it can affect the sex ratio of offspring by programing embryos to develop into males [17]. In *S. paramamosain*, MF induced lipid accumulation in the hepatopancreas to prepare for ovarian development, and orally administered MF boosted the production of the mature ovary [18,20]. Additionally, it played a role in inducing the molting of *T. schirnerae* [19]. Therefore, we suggest that the different expression profiles in the hepatopancreas and ovary may be closely related to the multiple functions of MF.

*MFE*-related functional studies are relatively scarce. In insects, the knockdown of *MFE* in *E. chinensis* resulted in a significant decrease in cantharidin content [27], and the knockdown of *CYP15A1* in *T. castaneum* at the end of the larval stage did not lead to precocious pupation [26]. However, a role of *MFE* in reproduction-related functions has not been reported in insects or crustaceans. We used RNAi technology for the first time to investigate the role of *Mn-MFE* in ovarian development. After RNAi, the percentage of ovaries developing past stages O-III and the GSI were lower in the experimental group than in the control group. At the end of the experiment, paraffin tissue sections of the ovaries revealed that only the developmental rate was affected, and there was no abnormal change in the subcellular structure. These results suggest that RNAi only promoted ovarian development and did not lead to structural aberrations. We also found that RNAi did not have a significant effect on molting. In previous functional studies of other genes in the insect hormone biosynthesis signaling pathway, we found that knocking down these genes not only inhibited ovarian development, but it also inhibited molting [37,44]. Although both *MFE* and the above enzymes in this study are in the insect hormone biosynthesis signaling pathway, their functions have been significantly changed, and *Mn-MFE* does not have the function of promoting molting.

In our previous study, we performed a comparative transcriptomic analysis of the hepatopancreas at five ovarian stages and identified many differentially expressed genes related to ovarian maturation. The *MFE* gene is one of these genes. The main purpose of this study was to investigate the role of this gene in the ovarian maturation of *M. nipponense*. However, this was a preliminary study conducted under experimental conditions. In practical aquaculture applications, environmental factors such as temperature, water quality, diet, and different developmental stages could significantly influence ovary development. The complex polygenic nature of traits can involve genes and environmental interactions. In future studies, we will include environmental factors and different development stages in functional analyses of *Mn-MFE* to obtain a better understanding of the role of this gene in the rapid sexual maturity of *M. nipponense*. MFE can epoxidize MF to JH and MF is multi-functional in crustaceans, playing roles in sex differentiation [17], lipid accumulation [18], molting [19], and promoting ovarian maturation [20]. In this study, we used RNAi to investigate the role of *Mn-MFE* only in female ovary maturation. In future steps, we will explore the role of *Mn-MFE* in males as well as its other functions in the hepatopancreas, and results will shed light on the distinct roles of this gene in sexual dimorphism, which can affect the social dominance of the species [45].

## 4. Materials and Methods

### 4.1. Experimental Animal

The female prawns used in the experiment came from the Dapu Scientific Experiment Base of the Freshwater Fisheries Research Center from May to August 2023. This study did not involve endangered or protected species and was approved by the Institutional Animal Protection and Use Ethics Committee of the Freshwater Fisheries Research Center of the Chinese Academy of Fishery Sciences (Wuxi, China). Female prawns were cultured in an indoor recirculating aquaculture system (recirculating glass aquarium tanks with a volume of about 100 L), and the water temperature was maintained at 24 ± 1 °C. The pH was 6.7 to 7.1, the dissolved oxygen content was 5.5 to 6.9 mg/L, and the ammonia nitrogen content was 0.25 to 0.28 mg/L. The prawns were fed with paludina in the morning and evening, with a feeding amount of about 3–5% of the culture weight. The culture water was cleaned every day.

### 4.2. Sample Collection

We used 30 healthy and viable adult female *M. nipponense* weighing 1.57 ± 0.24 g for tissue expression studies. We collected the following tissues for analysis: eyestalk (E), cerebral ganglion (Cg), heart (H), hepatopancreas (He), gills (G), ovary (O), and muscle (M). Each tissue sample was composed of 5 biological replicates and one replicate contained 3 randomly selected prawns. Fifty adult female *M. nipponense* (BW ± SD: 1.81 ± 0.39 g) at different ovarian periods were taken for ovarian expression pattern studies. Each sample of each stage contained 5 biological replicates and one replicate contained 3 randomly selected prawns. In our previous study [46], ovarian staging was defined as 5 stages based on ovarian color. Each stage sample contained 5 biological replicates and each replicate contained 20 randomly selected embryos. We sampled the larval (L) and post-larval (PL) developmental stages every 5 days, for a total of 9 time points: L1, L5, L10, L15, PL1, PL5, PL10, PL15, and PL25. Each stage sample contained 5 biological replicates and each replicate contained 10 randomly selected larvae. All samples were stored in liquid nitrogen prior to analysis.

### 4.3. RNA Extraction, cDNA Synthesis, and Sequence Verification

Total RNA was extracted from *M. nipponense* using the RNAiso Plus kit (TaKaRa, Shiga, Japan), the RNA concentration was determined, and the RNA quality was detected using gel electrophoresis. According to the manufacturer’s instructions, we converted single-stranded RNA to single-stranded cDNA using the M-MLV reverse transcriptase kit (TaKaRa). Partial *Mn-MFE* cDNA sequences were obtained from the hepatopancreas proteome of *M. nipponense* (http://proteomecentral.proteomexchange.org, PXD037141, accessed on 10 April 2023) [47]. Primers were designed for sequence validation using Primer 5.0 [48] (Table 1). The complete sequence was amplified by 3′ and 5′ cloning using a RACE PCR kit (TaKaRa). The PCR products were sequenced by Sangon Biotech (Shanghai, China).

### 4.4. Sequence Analysis

Open reading frame (ORF) analysis and amino acid sequence prediction of *Mn-MFE* full-length cDNA sequences were performed using the NCBI online analysis tool-ORF Finder (http://www.ncbi.nlm.nih.gov/gorf/gorf.html) (accessed on 11 April 2023). The similarity and consistency of ORFs were tested by means of BLAST comparison. Based on the predicted amino acid sequences, the molecular weight (MW) and isoelectric point (PI) of Mn-MFE proteins were estimated using the Protparam program in ExPasy (https://web.expasy.org/protparam/) (accessed on 11 April 2023). We used Signal 3.0 (https://www.novopro.cn/tools/signalp) (accessed on 11 April 2023) for signal peptide prediction and TMpred online software (https://dtu.biolib.com/DeepTMHMM) (accessed on 11 April 2023) to predict the transmembrane structural domains of amino acids. The functional structural domains of the Mn-MFE amino acid sequence were predicted using the PROSITE SCAN tool (https://www.ebi.ac.uk/interpro/) (accessed on 11 April 2023). Similar amino acid sequences were searched using BLASTP in GenBank (http://blast.ncbi.nlm.nih.gov/Blast.cgi) (accessed on 11 April 2023). We used DNAMAN 6.0 software to analyze the similarity between the amino acid sequences of MEF from different species [49], and we also used the ClustalW program in MEGA 7.0 to compare these sequences. We constructed the phylogenetic tree using MEGA 7.0 software. The calculation was repeated 1000 times by the bootstrap method and the neighbor-joining algorithm [50].

### 4.5. In Situ Hybridization (ISH)

We used ISH to detect the subcellular localization of *Mn-MFE* during five periods of ovarian development. One healthy adult female *M. nipponense* from each of 5 ovary development stages was selected, and the ovarian tissue samples were dissected and soaked with 4% paraformaldehyde buffer. The specific experimental steps of ISH were performed following the instruction manual for the CISH technique [44]. Both sense and antisense probes with digoxin signals were synthesized by Sangon Biotech (Table 1). The ISH of the sense probe sequence ATGTTGATCGTTGTTGCCTTGGTACTCCTGG and the antisense probe sequence CCAGGAGTACCAAGGCAACAACGATCAACAT was performed. HE was the blank control group, which was stained with hematoxylin-eosin. The sense probe was used for the experimental group. The antisense probe was used for the negative control group. All slides were examined using a light microscope.

### 4.6. RNA Interference (RNAi)

For RNAi analysis, we used the green fluorescent protein (*GFP*) gene as the control gene [51,52]. Interference primers for *Mn-MFE* and *GFP* were designed using the online website Snap Dragon (http://www.flyrnai.org/cgi-bin/RNAi_find_primers.pl) (Table 1). According to the instructions, double-stranded RNA (dsRNA) was synthesized in vitro using the Transcript AidTMT7 High Yield Transcription Kit (Fermentas, Inc., Rockville, MD, USA). We measured the concentration of dsRNA at 260 nm using the BioPhotometer Ultraviolet Spectrophotometer (Eppendorf, Hamburg, Germany). The dsRNA was diluted to 4 μg/g with DEPC water, and then the quality of dsRNA was assessed by means of agarose gel electrophoresis, followed by storage in an ultra-low-temperature refrigerator at −80 °C.

At the beginning of the interference experiment, the ovaries of most prawns were in the O-I stage, so we selected female prawns with ovaries in this stage as the experimental object. The breeding time of the short-term interference efficiency test was 7 days. We selected 90 healthy and viable female prawns (1.45 ± 0.16 g) at the O-I stage and randomly assigned them to the experimental or control groups, with 3 parallel (n = 15) in each group. The water temperature was maintained at 24 ± 1 °C, and the prawns were fed once in the morning and once in the evening. The control group and the experimental group were injected with ds-GFP and ds-MFE at a dose of 4 μg/g b.w. (body weight) into the pericardial cavity [46]. At 1, 4, and 7 days after injection, 9 prawns (3 biological replicates) were randomly collected from each group, and their ovaries were dissected and immediately frozen in liquid nitrogen for the analysis of interference efficiency.

The long-term interference experiment lasted 25 days. We selected 180 healthy and active female prawns (1.62 ± 0.25 g) at the O-I stage of ovarian development and randomly assigned them to the experimental or control groups, with 3 parallel (n = 30) in each group. The water temperature was maintained at 24 ± 1 °C, and the prawns were fed once in the morning and once in the evening. Once every 5 days, prawns in the control group and the experimental group were injected with ds-GFP and ds-MFE, respectively, at a dose of 4 μg/g b.w. (body weight). At days 1, 7, 14, and 21 after injection, 9 prawns (3 biological replicates) were randomly collected from each group. Ovaries were dissected, and weight and gonadal weight were recorded to determine GSI (GSI = gonadal weight/body weight × 100%). Ovarian samples were then immediately placed in liquid nitrogen for freezing and stored at −80 °C. Ovarian development was observed daily, focusing on female prawn past stage O-III (yolk accumulation stage), and the ovarian development period of each individual was recorded. We also counted the number of molting individuals in each replicate every day and used the data to calculate the cumulative molting frequency. At the end of the experiment, we randomly selected one prawn at stage O-II from each of the experimental and control groups, dissected its ovary, and used it for paraffin tissue sectioning.

### 4.7. Quantitative PCR (qPCR)

We used the *EIF* gene as the internal reference gene and formulated the reaction system for qPCR following a previous study [46]. We calculated the relative expression of genes using the 2^−ΔΔCt^ method [53]. All quantitative data conformed to the homogeneity of variance and the normal distribution, and all data were presented as mean ± standard deviation. The data were then statistically analyzed with one-way analysis of variance and two-tailed *t*-tests using SPSS 20.0 software [54]. *p* < 0.05 indicated significant difference, and *p* < 0.01 indicated highly significant difference.

## 5. Conclusions

This is the first study of the reproductive function of *methyl farnesoate epoxidase* (*MFE*) in crustaceans (Genbank accession No.: PP582172). The sequence analysis showed that the structural domain FGCG of *M. nipponense* has nine more amino acids than that of the other species analyzed, suggesting that its functions in this species may be significantly different from those in other species. *Mn-MFE* expression remains stable during the larval period and plays a critical role mainly during gonadal differentiation. We found that the expression of *Mn-MFE* in the ovary was significantly positively correlated with ovarian development and that its expression in the hepatopancreas was negatively correlated with ovarian development. RNAi experiments confirmed that *Mn-MFE* plays an important role in ovarian development. The experimental results of this study provide a basis for related studies of crustacean ovarian development.

## Figures and Tables

**Figure 1 ijms-25-07318-f001:**
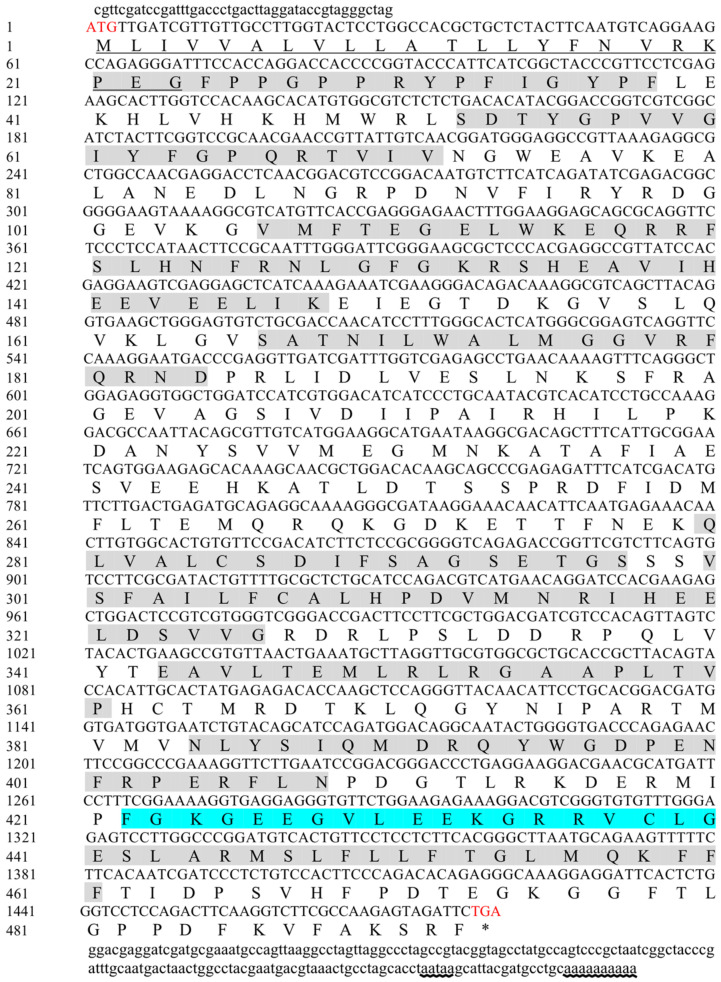
Nucleotide and deduced amino acid sequences of Mn-MFE. The start codon ATG and the stop codon TAG are marked in red. The stop codon TAG in the amino acid sequence is represented by an asterisk (*). The first 23 amino acids of the signal peptide are underlined in black. The gray shaded region is the functional structural domain of Mn-MFE. The blue shaded region indicates the FGCG structural domain. The tailing signal and poly(A) are indicated by wavy underlining.

**Figure 2 ijms-25-07318-f002:**
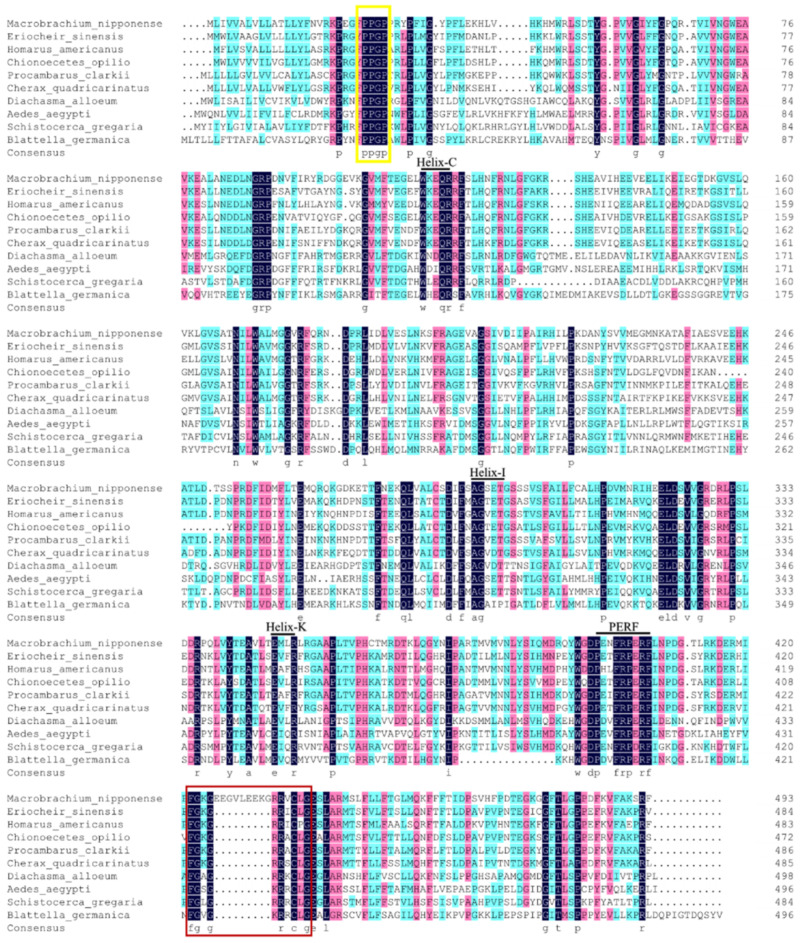
Multiple sequence comparison of amino acid sequences of Mn-MFE with those of other species. Sequence identities are indicated in black and conserved variants in pink. The black lines mark the four conserved structural domain sites in sequence. Boxed in yellow is the typical “PPGP” hinge of the P450 family. The signature heme-binding region (FGCG) is boxed in red. MFEs: *Eriocheir sinensis*, XP_050712089.1, *Homarus americanus*, XP_042208946.1, *Chionoecetes opilio*, KAG0710129.1, *Procambarus clarkia*, XP_045594864.1, *Cherax quadricarinatus*, XP_053649007.1, *Diachasma alloeum*, XP_015114454.1, *Aedes aegypti*, XP_021697821.1, *Schistocerca gregaria*, ADV17351.1, *Blattella germanica*, PSN47977.1.

**Figure 3 ijms-25-07318-f003:**
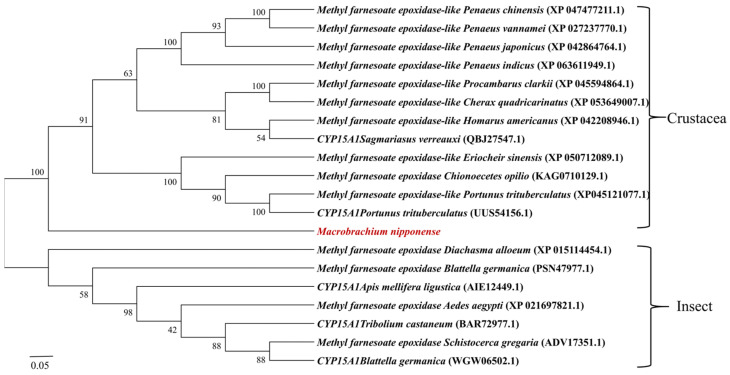
Phylogenetic evolutionary tree of *Mn-MFE*. The graph was generated by the MEGA 7.0 program using the adjacency method. The numbers on the branch represent the bootstrap percentages of the phylogenetic tree. Bootstrap copy to 1000. The numbers in brackets indicate the GenBank accession numbers.

**Figure 4 ijms-25-07318-f004:**
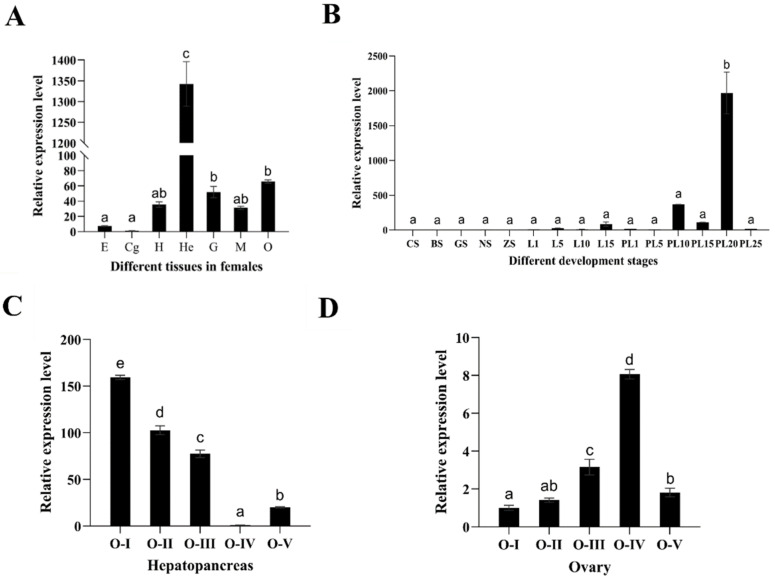
Expression analysis of *Mn-MFE.* (**A**) Expression patterns in different tissues (five biological replicates, three prawns/replicate). (**B**) Expression patterns at different developmental stages (five biological replicates, 20 embryos/replicate) (five biological replicates, 10 larvae/replicate). (**C**) Expression patterns in hepatopancreas at different ovarian developmental stages (five biological replicates, three prawns/replicate). (**D**) Expression patterns in ovaries at different ovarian developmental stages (five biological replicates, three prawns/replicate). (**A**) E, eyestalk; Cg, cerebral ganglion; H, heart; He, hepatopancreas; G, gill; M, muscle; O, ovary. (**B**) CS, cleavage stage; BS, blastocyst stage; GS, gastrulation stage; NS, nauplius stage; ZS, zoea stage; L1, larvae at day 1 after hatching; L5, larvae at day 5 after hatching; L10, larvae at day 10 after hatching; L15, larvae at day 15 after hatching; PL1, late larval stage of the 1st day; PL5, late stage of the 5th day; PL10, late stage of the 10th day; PL15, late stage of the 15th day; PL20, late stage of the 20th day; PL25, late stage of the 25th day. (**C**,**D**) O-I, undeveloped stage; O-II, developing stage; O-III, nearly ripe stage; O-IV, ripe stage; and O-V, spent stage. Data are presented as the mean ± standard. Different lowercase letters indicate significant differences (*p* < 0.05).

**Figure 5 ijms-25-07318-f005:**
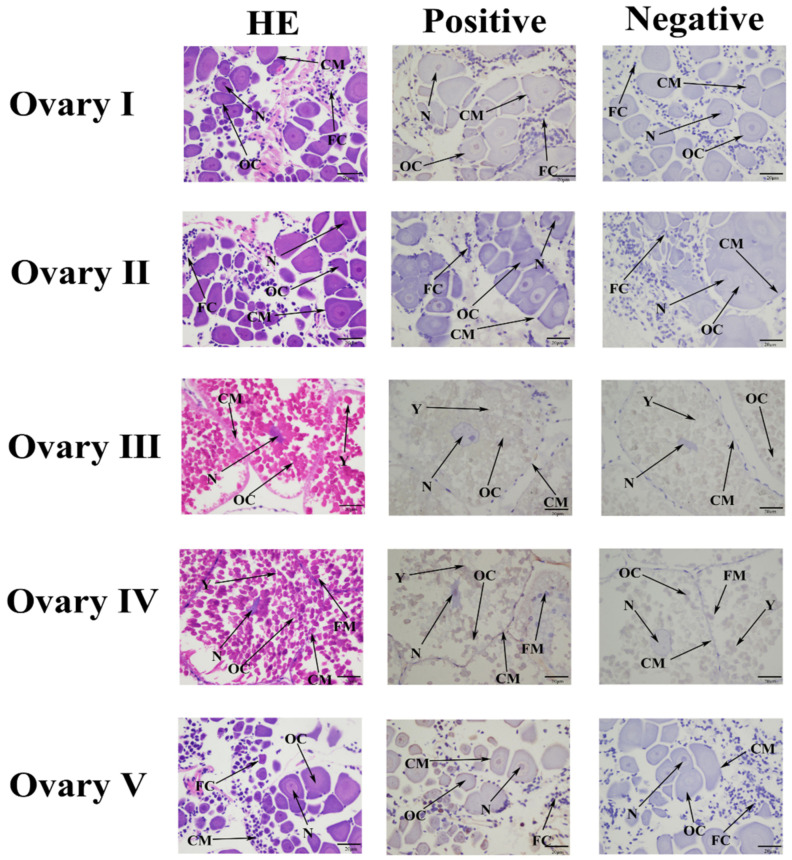
Location of *Mn-MFE* in ovary detected by in situ hybridization. HE: blank control group; Positive: experimental group; Negative: control group. The brown color in the positive figure represents the digoxin-labeled gene expression signal compared to the negative figure. OC: oocyte; N: nucleus; CM: cell membrane; Y: yolk granule; FC: follicular cells; FM: follicle membrane. Scale bars: ×400 (20 μm).

**Figure 6 ijms-25-07318-f006:**
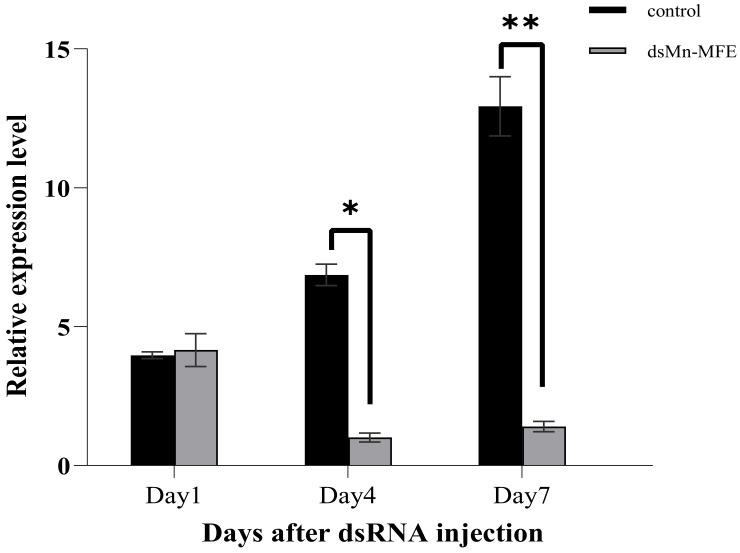
Expression levels in ovaries after dsRNA injection (three biological replicates, three prawns/replicate). Data are presented as the mean ± standard. * indicates significant difference (*p* < 0.05), ** indicates highly significant difference (*p* < 0.01).

**Figure 7 ijms-25-07318-f007:**
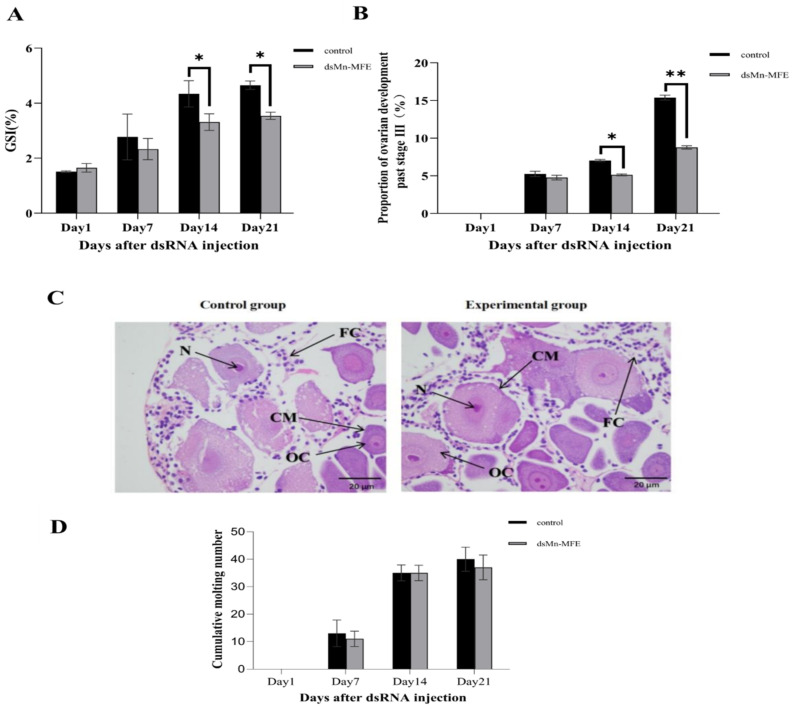
Function of *Mn-MFE* in the ovary. (**A**) Changes in GSI (%) of female *Macrobrachium nipponense* after RNAi (three biological replicates, three prawns/replicate). (**B**) Changes in the cumulative proportion of female *M. nipponense* ovaries over stage O-III after RNAi. (**C**) Observational analysis of paraffin sections of ovaries from two groups after RNAi. (**D**) Changes in the cumulative number of molts in female *M. nipponense* after RNAi. Data are presented as the mean ± standard. * denotes significant difference (*p* < 0.05), ** denotes highly significant difference (*p* < 0.01). OC: oocyte; N: nucleus; CM: cell membrane; FC: follicular cells. Scale bars: ×100.

**Table 1 ijms-25-07318-t001:** Primers used in this study.

Primer	Primer Sequence (5′-3′)
*Mn-MFE* F (ORF)	CTACTTCAATGTCAGGAAGCCAG
*Mn-MFE* R (ORF)	GACAGAGGGATCGATTGTGAAGA
*Mn-MFE* R (5′)	AACGGGTAGCCGATGAATGGGTA
*Mn-MFE* F (3′)	GCCCGGATGTCACTGTTCCTCCT
*Mn-MFE* F (qPCR)	TGATCGTTGTTGCCTTGGTACTC
*Mn-MFE* R (qPCR)	ACCAAGTGCTTCTCGAGGAACGG
*EIF*-F (qPCR)	CATGGATGTACCTGTGGTGAAAC
*EIF*-R (qPCR)	CTGTCAGCAGAAGGTCCTCATTA
*Mn-MFE* probe	ATGTTGATCGTTGTTGCCTTGGTACTCCTGG
*Mn-MFE* anti-probe	CCAGGAGTACCAAGGCAACAACGATCAACAT
*Mn-MFE* iF (RNAi)	TAATACGACTCACTATAGGGTGATCGTTGTTGCCTTGGTACTC
*Mn-MFE* iR (RNAi)	TAATACGACTCACTATAGGGACCAAGTGCTTCTCGAGGAACGG
*GFP* iF (RNAi)	GATCACTAATACGACTCACTATAGGGTCCTGGTCGAGCTGGACGG
*GFP* iR (RNAi)	GATCACTAATACGACTCACTATAGGGCGCTTCTCGTTGGGGTCTTTG

## Data Availability

The original contributions presented in the study are included in the article, further inquiries can be directed to the corresponding authors.

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
