# Peer review of "Functional Study of the Role of the Methyl Farnesoate Epoxidase Gene in the Ovarian Development of Macrobrachium nipponense"

_ijms, 2024, doi:10.3390/ijms25137318_

Round 1

Reviewer 1 Report

Comments and Suggestions for Authors

This manuscript titled "Functional study on Methyl farnesoate epoxidase gene in ovarian development of Macrobrachium nipponense" (IJMS 2996350) successfully cloned the full-length cDNA of MFE gene and investigated the molecular characterizations, evolutionary pattern, expression patterns, and reproductive function of MFE gene in Macrobrachium nipponense. Data from this study could highlight its biological function in ovarian development and provide the necessary foundation for further investigating molecular mechanism of ovarian development in Macrobrachium nipponense and other crustaceans.

Core contents of this manuscript are important for the evolution and reproductive development in Macrobrachium nipponense. In this case, the paper should be proofread throughout by a native speaker to avoid grammatical errors and improve its readability of whole paper, particularly the description in the section of "Results". 

Major comments:

1. In the section of "1.Introduction", the cited references are missing from several sentences. For example, in Line 60-62 and Line 67-70 lack of the citation of previous reports. Please add the relevant reference information in the corresponding part. 

2. Regarding the "2. Results" section, several descriptions are not concise. For example, Line 86-87 and Line 135-136 contain the redundant statements of results. There are similar errors in the in the other text of result part. It is suggested that the authors shorten or remove the redundant content for a clear and brief summary of main results in this study. 

3. In the section of "Discussion", several statements contain the unintentional duplicate result description. So, the relevant part should be rephrased and modified with emphasis for better clarifying your findings in this study. Additionally, some references should be correctly cited in this part and replaced with latest published literatures (2019-2024). 

4. In Line 321-322, only water temperature is included. What about the other parameters of water quality, including pH, dissolved oxygen, total ammonia, and so on? The authors should provide more information on aquatic environmental parameters to "4.1. Experimental animal" section. 

5. The authors should check the references carefully according to the Instructions for Authors. Multiple are present in the Reference list, with many page numbers missing, along with other inconsistencies like abbreviated vs. full journal names, capitalized vs. lower-case article titles, italicized vs. non-italic latin name of species. For example, in Reference 3, page number information should be provided. In Reference 1, the title is written with capital letters for the first letter of all words, but in most other references, the words are written with capital initial letter of the first word. There were similar errors in the other references.

Additionally, the cited literatures were a little old, only 9 references (total references: 42) published in 2019-2024. For example, Reference 24 is published in 1961. Please make sure 40%-50% of the references are within 5 years (2019-2024). 

Minor comments:

1. In Line 39, replace ".[4]" with "[4].". 

2. Figure 5 lacks the interpretation of some letters ("Y" and "FM") in the legend of Fig.5. What do these letters represent? The authors should provide more information in the legend of Fig.5 in the revised version. 

Other errors were presented in the PDF file.

Therefore, this manuscript will be accepted after minor revision.

Comments on the Quality of English Language

The core content of this manuscript (IJMS 2996350) entitled "Functional study on Methyl farnesoate epoxidase gene in ovarian development of Macrobrachium nipponense" is valuable for the evolution and reproductive development in Macrobrachium nipponense and other crustaceans. But there are some minor mistakes, including the references format. Additionally, some descriptions in the section of "Results" should be streamlined or directly removed. It is recommended that the whole paper should be proofread carefully to avoid any typos and grammatical errors in the revised manuscript.

Author Response

Dear Reviewer 1,

First of all, we are very grateful for your recognition of our research work. Thank you very much for your comments and suggestions. The valuable comments from you not only helped us with the improvement of our manuscript, but suggested some ideas for future studies.

Below you will find our responses to your comments:

Major comments:

  1. In the section of "1. Introduction ", the cited references are missing from several sentences. For example, in Line 60-62 and Line 67-70 lack of the citation of previous reports. Please add the relevant reference information in the corresponding part.

Response: Thanks for your kindly reminding. We add the missed reference in the revised manuscript.

Minakuchi, C.; Ishii, F.; Washidu, Y.; Ichikawa, A.; Tanaka, T.; Miura, K.; Shinoda, T. Expressional and functional analysis of CYP15A1, a juvenile hormone epoxidase, in the red flour beetle Tribolium castaneumJ. Insect Physiol. 201580, 61-70.

Jiang, M.; Lü, S.; Zhang, Y. Characterization of juvenile hormone related genes regulating cantharidin biosynthesis in Epicauta chinensis. Sci. Rep. 2017, 7, 2308.

  1. Regarding the "2. Results" section, several descriptions are not concise. For example, Line 86-87 and Line 135-136 contain the redundant statements of results. There are similar errors in the in the other text of result part. It is suggested that the authors shorten or remove the redundant content for a clear and brief summary of main results in this study.

Response: Thanks for your kind comment. We corrected the many errors and adjusted according to your advice in the revised manuscript. 

On line 86-87, we revised as follows: “The open reading frame (ORF) of Mn-MFE was validated using PCR sequencing (Fig. 1). The full-length Mn-MFE cDNA sequence was 1695 bp, and the ORF length was 1482 bp (Genbank accession No.: PP582172).”

On line 135-136, we revised as follows: “Results as shown in Figure 4, the expression level of Mn-MFE was highest in the hepatopancreas (P < 0.05), followed by the ovary and gill. It was weakly expressed in heart and muscle tissue and was hardly expressed in the eyestalk and cerebral ganglion.”

  1. In the section of "Discussion", several statements contain the unintentional duplicate result description. So, the relevant part should be rephrased and modified with emphasis for better clarifying your findings in this study. Additionally, some references should be correctly cited in this part and replaced with latest published literatures (2019-2024).

Response: Thanks for your kind comment. We have revised the unintentional duplicate result description. And we carefully checked and updated the relevant references in the revised manuscript. 

  1. In Line 321-322, only water temperature is included. What about the other parameters of water quality, including pH, dissolved oxygen, total ammonia, and so on? The authors should provide more information on aquatic environmental parameters to "4.1. Experimental animal" section.

Response: Thanks for your good suggestion. We have added more information on aquatic environmental parameters in the section “4.1. Experimental Animals” as follows: The pH was 6.7 to 7.1, the dissolved oxygen content was 5.5 to 6.9 mg/L, and the ammonia nitrogen content was 0.25 to 0.28 mg/L.

  1. The authors should check the references carefully according to the Instructions for Authors. Multiple are present in the Reference list, with many page numbers missing, along with other inconsistencies like abbreviated vs. full journal names, capitalized vs. lower-case article titles, italicized vs. non-italic latin name of species. For example, in Reference 3, page number information should be provided. In Reference 1, the title is written with capital letters for the first letter of all words, but in most other references, the words are written with capital initial letter of the first word. There were similar errors in the other references.

Response: Thanks for your valuable comment. We have carefully checked the relevant references and have revised and corrected the errors.

Additionally, the cited literatures were a little old, only 9 references (total references: 42) published in 2019-2024. For example, Reference 24 is published in 1961. Please make sure 40%-50% of the references are within 5 years (2019-2024).

Response: Thanks for your valuable comment.We updated the references so that 40%-50% of the references were updated within 5 years (2019-2024).

Minor comments:

  1. In Line 39, replace ".[4]" with "[4].".

Response: Thank you for your kind reminder, it has been replaced.

  1. nipponensereaches sexual maturity rapidly. During the breeding season (April-October), hatching to ovarian development and maturity occurs within about 45 days [4].
  2. Figure 5 lacks the interpretation of some letters ("Y" and "FM") in the legend of Fig.5. What do these letters represent? The authors should provide more information in the legend of Fig.5 in the revised version. 

Response: Thank you for your valuable comment. We have explained "Y" and "FM" in the revised version of the legend in Figure 5.

Figure 5. Location of Mn-MFE in ovary detected by in situ hybridization. HE: blank control group; Positive: experimental group; Negative: control group. The brown color in the positive figure represents the digoxin-labeled gene expression signal compared to the negative figure. OC: Oocyte; N: nucleus; CM: Cell membrane; Y: yolk granule; FC: follicular cells; FM: follicle membrane. Scale bars: ×400.

Reviewer 2 Report

Comments and Suggestions for Authors

The article shows the importance of of MF in ovarian development of the Oriental River Prawn. This study is very important for the understanding of crustacean reproductive physiology.

Major:

·       Introduction: lots of scientific names for species are mentioned in this section. It is advised that also the common name will be mentioned as well. For example: The German cockroach or the cockroach (Blattella germanica) instead of Blattella germanica only.

·       Line 77: The stages of ovarian development should be elaborated. What each number represents. It is mentioned in line 171 but I think it should come earlier in the text.

·       Line 82: The problem of early sexual maturation should be explained here.

·       Figure 5 is not clear. The expression or lack of expression should be highlighted better than it is now. Also, the abbreviation of HE and explanation of “positive” and “negative” should be in the figure caption.

·       RNAi experiment, Figures 6 and 7 and Lines 184-222: if all the % are means, also SD should be mentioned as well as number of animals in each group. That’s including both text and figure captions. Figure 7B is not clear – what was measured and why. Figure 7C – if no change is observed, how do you explain the difference in GSI? If the highest expression is in the hepatopancreas – why the RNAi analysis was not performed on the hepatopancreas in addition to the ovary?

·       Line 294: so what is the possible reason for this negative correlation between hepato and ovary? It is important to discuss it rather than only mentioning other studies.

·       I think the fact that the highest expression is in the hepato but the RNAi analysis was done on the ovary only should be explained. Also, the authors may want to suggest future experiment on the hepato.

Minor:

·       Name of species should be in italic throughout the entire manuscript including the title and figures.

·       Line 13: JH should not be abbreviated here since it’s the first mention in the article.

·       Line 107: reference for the software is missing.

·       Figure 3: spaces should be added between the text and the parenthesis on each branch. Also, more details are required for the caption (what are the numbers on the branch, any bootstrapping? If so – how many? What are the names in parenthesis? Genebank accession numbers?).

·       Line 136: “as” instead of “As”

·       Line 187: “showed in Figure 6”

·       Line 188: “showed almost no difference” instead of "was almost no difference”

·       Line 315: “prawns” instead of “prawn”.

·       Line 348: reference is missing for Primer 5.0.

·       Line 368: What about bootstrapping?

·       Lines 352-368: references for all software are required.

·       Lines 374-379: it is not clear what is the experimental, control and blank groups for in-situ here.

·       Line 376: how many females were examined until you choose the representative photos?

·       Line 379: a reference to the protocol is required.

·       Lines 396 and 407: “with ds-GFP and ds-MFE, respectively”

·       Line 416: how many shrimp were randomly selected?

·       Line 426: full name of the gene and accession number are required here.

·       Section 4.7 – qPCR – the details about the statistical analysis should be written here. What test were done? Also a reference for SPSS is required.

Comments on the Quality of English Language

Generally okay but moderate editing is required. Assistance from a professional linguistic editor is suggested.

Author Response

Dear Reviewer 2,

First of all, we are very grateful for your recognition of our research work. Thank you very much for your comments and suggestions. The valuable comments from you not only helped us with the improvement of our manuscript, but suggested some ideas for future studies.

Below you will find our responses to your comments:

Major:

  1.  Introduction: lots of scientific names for species are mentioned in this section. It is advised that also the common name will be mentioned as well. For example: The German cockroach or the cockroach (Blattella germanica) instead of Blattella germanica only.

 Response: Thanks for your kind suggestion. We have added the common name as follows.

The MFE gene has been found in desert locust (Schistocerca gregaria) [23], German cockroach (Blattella germanica) [24], Pacific beetle cockroach (Diploptera punctata) [25], red flour beetle (Tribolium castaneum) [26] and other species.

  1. Line 77: The stages of ovarian development should be elaborated. What each number represents. It is mentioned in line 171 but I think it should come earlier in the text.

Response: Thanks for your kind suggestion, we have made changes as follows.

We used in situ hybridization (ISH) to detect the subcellular localization of Mn-MFE during the following five periods of ovarian development: O-I, undeveloped stage; O-II, developing stage; O-III, nearly-ripe stage; O-IV, ripe stage; and O-V, spent stage.

  1. Line 82: The problem of early sexual maturationshould be explained here.

Response: Thanks for your kind suggestion. Based on your suggestion, we have further explained the early sexual maturation of M. nipponense on line 82 in the revised version.

  1. Figure 5 is not clear. The expression or lack of expression should be highlighted better than it is now. Also, the abbreviation of HE and explanation of “positive” and “negative” should be in the figure caption.

Response: Thanks for your kind suggestions. We have added to the figure note: “The brown color in the positive figure represents the digoxin-labeled gene expression signal compared to the negative figure.” Because the expression of Mn-MFE in the ovary is relatively low, the signal expression may not be strong, so it looks pale.

We have added omissions to Figure 5 in the revised original manuscript: “Figure 5. Location of Mn-MFE in ovary detected by in situ hybridization. HE: blank control group; Positive: experimental group; Negative: control group. The brown color in the positive figure represents the digoxin-labeled gene expression signal compared to the negative figure. OC: Oocyte; N: nucleus; CM: Cell membrane; Y: yolk granule; FC: follicular cells; FM: follicle membrane. Scale bars: ×400.”

  1. RNAi experiment, Figures 6 and 7 and Lines 184-222: if all the % are means, also SD should be mentioned as well as number of animals in each group. That’s including both text and figure captions. Figure 7B is not clear – what was measured and why. Figure 7C – if no change is observed, how do you explain the difference in GSI? If the highest expression is in the hepatopancreas – why the RNAi analysis was not performed on the hepatopancreas in addition to the ovary?

Response: Thanks for your kind suggestion. We added the number of animals in each group and added the relevant questions in the notes to Figures 6 and 7, and the data are presented as mean ± standard deviation.

Figure 7B represents the changes in the cumulative proportion of female Macrobrachium nipponense ovary past stages O-III after RNAi. At the beginning of the experiment, the ovaries of most female prawn were in stage O-I. Stage III of the ovary was the accumulation of yolk and the beginning of the accelerated development stage. The color and volume of the ovary changed significantly, which was a sign of gradual maturation of the ovary. The effect of Mn-MFE on ovarian maturation in female Macrobrachium nipponense is more intuitively reflected by comparison with the control group.

Gonadosomatic index (Figure. 7A) was determined by randomly collecting 9 prawns from each group (3 biological replicates), dissecting out the ovaries, and recording body weight and gonad removal weight to determine GSI. Ovarian development is uncertain, so the range of changes is large. Sections (Figure. 7C) were taken at the end of the experiment and randomly selected prawn at the same period (O-II) were tested for structural changes after interference. Therefore, the magnitude of change was small.

In our previous study, we performed comparative transcriptomic analysis of the hepatopancreas at five ovarian stages and identified many differentially expressed genes related to ovarian maturation. The MFE gene is one of these genes. The main purpose of this study was to investigate the role of this gene in ovarian maturation of M. nipponense.

  1. Line 294: so what is the possible reason for this negative correlation between hepato and ovary? It is important to discuss it rather than only mentioning other studies.

Response: Thanks for your kind suggestion. The expression of Mn-MFE in hepatopancreas and ovary at 5 stages were further studied. The results showed that the expression of Mn-MFE in the ovary was significantly positively correlated with ovarian development, while it was not the same in the hepatopancreas. There are no reports of MFE being involved in ovarian development in insects. However, MF has been reported to be multi-functional in crustaceans. In Daphnia, it can affect the sex ratio of offspring by programing embryos to develop into males (Olmstead et al. 2002). In S. paramamosain, MF induced lipid accumulation in hepatopancreas to prepare for ovarian development, and orally administered MF boosted the production of the mature ovary (Fu et al. 2022; Muhd-Farouk et al. 2019). Additionally, it played a role in inducing molting of T. schirnerae (Raghavan et al. 2019). Therefore, we suggest that the different expression profiles in the hepatopancreas and ovary may be closely related to the multiple functions of MF.

Olmstead, A.W.; Leblanc, G.A. Juvenoid hormone methyl farnesoate is a sex determinant in the crustacean Daphnia magna. J. Exp. Zoo. 2002, 293, 736-739.

Fu, Y.; Zhang, F.Y.; Ma, C.Y. Wang, W. Liu, Z.Q.; Chen, W. Zhao, M.; Ma, L.B. Comparative metabolomics and lipidomics of four juvenoids application to Scylla paramamosain hepatopancreas: implications of lipid metabolism during ovarian maturation. Front. Endocrinol. 2022, 13, 886351.

Muhd-Farouk, H.; Nurul, H.A.; Abol-Munafi, A.B.; Mardhiyyah, M.P.; Hasyima-lsmail, N.; Manan, H.; Fatihah, S.N.; Amin-Safwan, A.; Lkhwanuddin, M. Development of ovarian maturations in orange mud crab, Scylla olivacea (Herbst, 1796) through induction of eyestalk ablation and methyl farnesoate. Arab J. Basic App. Sci. 201926, 171–181.

Raghavan, S.D.A.; Ayanath, A. Effect of 20-OH ecdysone and methyl farnesoate on moulting in the freshwater crab Travancoriana schirnerae. Invertebr. Reprod. Dev. 2019, 63, 309-318.

  1.  I think the fact that the highest expression is in the hepato but the RNAi analysis was done on the ovary only should be explained. Also, the authors may want to suggest future experiment on the hepato.

Response: Thanks for your kind suggestion. In our previous study, we performed comparative transcriptomic analysis of the hepatopancreas at 5 ovarian stages and identified many differentially expressed genes related to ovarian maturation. The MFE gene is one of these genes. The main purpose of this study was to investigate the role of this gene in ovarian maturation of M. nipponense. MFE can epoxidize MF to JH and that MF is multi-functional in crustaceans, playing roles in sex differentiation (Olmstead et al. 2002), lipid accumulation (Fu et al. 2022), molting (Raghavan et al. 2019), and pro-moting ovarian maturation (Muhd-Farouk et al. 2019). In this study, we used RNAi to investigate the role of Mn-MFE only in female ovary maturation. In future steps, we will explore the role of Mn-MFE in the hepatopancreas.

Olmstead, A.W.; Leblanc, G.A. Juvenoid hormone methyl farnesoate is a sex determinant in the crustacean Daphnia magna. J. Exp. Zoo. 2002, 293, 736-739.

Fu, Y.; Zhang, F.Y.; Ma, C.Y. Wang, W. Liu, Z.Q.; Chen, W. Zhao, M.; Ma, L.B. Comparative metabolomics and lipidomics of four juvenoids application to Scylla paramamosain hepatopancreas: implications of lipid metabolism during ovarian maturation. Front. Endocrinol. 2022, 13, 886351.

Raghavan, S.D.A.; Ayanath, A. Effect of 20-OH ecdysone and methyl farnesoate on moulting in the freshwater crab Travancoriana schirnerae. Invertebr. Reprod. Dev. 2019, 63, 309-318.

Muhd-Farouk, H.; Nurul, H.A.; Abol-Munafi, A.B.; Mardhiyyah, M.P.; Hasyima-lsmail, N.; Manan, H.; Fatihah, S.N.; Amin-Safwan, A.; Lkhwanuddin, M. Development of ovarian maturations in orange mud crab, Scylla olivacea (Herbst, 1796) through induction of eyestalk ablation and methyl farnesoate. Arab J. Basic App. Sci. 201926, 171–181.

Minor:

  1. Name of species should be in italic throughout the entire manuscript including the title and figures.

Response: Thanks for your kind reminder. We have made corresponding change in the revised manuscript.

  1. Line 13: JH should not be abbreviated here since it’s the first mention in the article.

Response: Thanks for your kind suggestions. We have made corresponding change in the revised manuscript.

Methyl farnesoate epoxidase (MFE) is a gene encoding an enzyme related to the last step of Juvenile hormone biosynthesis.

  1.  Line 107: reference for the software is missing.

Response: Thanks for your kind reminder. We have added the reference to software in the "4.4 Sequence Analysis" section of the revised manuscript.

We used DNAMAN 6.0 software to analyze the similarity between the amino acid sequences of MEF from different species (Ji et al. 2020).

Ji, L.Y.; Zhou, A.P.; Yu, X.Y.; Dong, Z.H.; Zhao, H.P.; Xue, H.; Wu, W.J. Differential expression analysis of the SRB1 gene in fluconazole-resistant and susceptible strains of Candida albicans. J. Antibiot.2020,73,309-313.

  1. Figure 3: spaces should be added between the text and the parenthesis on each branch. Also, more details are required for the caption (what are the numbers on the branch, any bootstrapping? If so – how many? What are the names in parenthesis? Genebank accession numbers?).

Response: Thanks for your kind reminder. We have added spaces between the text and parentheses in each branch. The numbers on the branch represent the bootstrap percentages of the phylogenetic tree, indicating the reliability of the branch, and we have added relevant explanations in its graph. Bootstrap copy to 1000.The numbers in brackets indicates the GenBank accession numbers.

  1. Line 136: “as” instead of “As”

Response: Thanks for your kind reminder. We have made the following change in the revised manuscript.

Results as shown in Figure 4, the expression level of Mn-MFE was highest in the hepatopancreas (P < 0.05), followed by the ovary and gill. It was weakly expressed in heart and muscle tissue and was hardly expressed in the eyestalk and cerebral ganglion.

  1. Line 187: “showed in Figure 6”

Response: Thanks for your kind suggestions. We have made following change in the revised manuscript as follows.

The results of short-term interference experiments showed in Figure 6.

  1. Line 188: “showed almost no difference” instead of "was almost no difference”

Response: Thanks for your kind reminder. We have made corresponding change in the revised manuscript.

On the 1st day after injection, the expression of Mn-MFE mRNA showed almost no difference between the two groups (P>0.05).

  1. Line 315: “prawns” instead of “prawn”.

Response: Thanks for your kind suggestion. We have made following change in the revised manuscript.

The female prawns used in the experiment came from the Dapu Scientific Experiment Base of the Freshwater Fisheries Research Center from May to August 2023.

  1. Line 348: reference is missing for Primer 5.0.

Response: Thanks for your kind reminder. We have added the missing reference to the revised manuscript.

Primers were designed for sequence validation using Primer 5.0 (Fang et al. 2024).

Fang, Z.Y.; Pazienza ,L.T.; Zhang, J.; Tam, C.P.; Szostak, J.W. Catalytic Metal Ion-Substrate Coordination during Nonenzymatic RNA Primer Extension. J. Am. Chem. Soc. 2024, 146, 10632-10639.

  1. Line 368: What about bootstrapping?

Response: Thanks for your kind reminder. Regarding line 368, we have added the following supplement: “We constructed the phylogenetic tree using MEGA 7.0 software. The calculation was repeated 1000 times by the bootstrap method and the neighbor-joining algorithm.”

  1. Lines 352-368: references for all software are required.

Response: Thanks for your kind suggestion. We have added the corresponding reference in the revised manuscript.

  1. Lines 374-379: it is not clear what is the experimental, control and blank groups for in-situ here.

Response: In lines 374-379, we have added the following: “HE was the blank control group, which was stained with hematoxylin-eosin. Positive experimental group used the sense probe. Antisense probe was used for the negative control group.”

  1. Line 376: how many females were examined until you choose the representative photos?

Response: One healthy adult female M. nipponense from each of 5 ovary development stages was selected, and the ovarian tissue samples were dissected and soaked with 4% paraform-aldehyde buffer.

  1. Line 379: a reference to the protocol is required.

Response: Thanks for your kind suggestions. We have added the reference to the protocol in the revised manuscript.

The specific experimental steps of ISH were performed following the instruction manual for the CISH technique (Wang et al. 2023).

Wang, J.; Jiang, S.; Zhang, W.; Xiong, Y.; Jin, S.; Cheng, D.; Zheng, Y.; Qiao, H.; Fu, H. Function Analysis of Cholesterol 7-Desaturase in Ovarian Maturation and Molting in Macrobrachium nipponense: Providing Evidence for Reproductive Molting Progress. Int. J. mol. sci. 2023, 24, 6940.

  1. Lines 396 and 407: “with ds-GFP and ds-MFE, respectively”

Response: Thanks for your kind reminder. We have made changes in the revised manuscript.

  1. Line 416: how many shrimp were randomly selected?

Response: At the end of the experiment, we randomly selected one prawn at stage O-II from each of the experimental and control groups, dissected its ovary, and used it for paraffin tissue sectioning.

  1. Line 426: full name of the gene and accession number are required here.

Response: Thanks for your kind suggestion. In line 426, we have added the relevant content to the revised manuscript: This is the first study of the reproductive function of methyl farnesoate epoxidase (MFE) in crustaceans (Genbank accession No.: PP582172).

  1. Section 4.7–qPCR–the details about the statistical analysis should be written here. What test were done? Also a reference for SPSS is required.

Response: Thanks for your kind suggestion. We have included detailed information on the statistical analysis in the revised manuscript. Details are as follows: “We calculated the relative expression of genes using the 2–ΔΔCt method. All quantitative data conformed to the homogeneity of variance and the normal distribution, and all data were presented as mean ± standard deviation. The data were then statistically analyzed by one-way analysis of variance and two-tailed t-tests using SPSS 20.0 software (Kang et al. 2018).” The reference to SPSS has also been added to the revised draft.

Kang, J.H.; Koo, S.M. Verification of Cervical Cancer Prevention Effect Using SPSS 20.0 WIN Program. Adv. Sci. Lett. 2018, 24, 2166-2170.

Reviewer 3 Report

Comments and Suggestions for Authors

The research article explores the function of the Methyl farnesoate epoxidase (MFE) gene in the ovarian development of Macrobrachium nipponense, a species of freshwater prawn. Using molecular biology techniques such as cDNA cloning, quantitative PCR, in situ hybridization, and RNA interference, the study demonstrates that MFE is highly expressed in the hepatopancreas and ovaries, and is crucial for ovarian maturation but does not affect molting. This investigation provides new insights into the reproductive regulation in crustaceans and offers potential molecular targets for controlling rapid sexual maturation in aquaculture, enhancing the sustainability of prawn farming. However, this manuscript requires major revisions in the following two areas before it can be considered for publication.

The article utilizes RNA interference (RNAi) to explore the function of the MFE gene, demonstrating its role in ovarian development. However, the control group design might not sufficiently exclude the impact of other potential factors. To enhance the rigor of the study, it is recommended to include additional control groups, such as using non-specific RNA sequences for negative controls and conducting identical experiments across different developmental stages or environmental conditions. This will help verify the repeatability and specificity of the results.

The manuscript mentions the use of qPCR and other biological assays, but the description of the statistical analysis methods is relatively brief in the methodology section. To improve transparency and verifiability, it is advisable to provide detailed information about the statistical tests used, significance levels (e.g., P<0.05), and the specific versions and settings of the data processing software. This not only aids in understanding the robustness of the results but also facilitates peer review and future replication studies.

Author Response

Dear Reviewer 3,

First of all, we are very grateful for your recognition of our research work. Thank you very much for your comments and suggestions. The valuable comments from you not only helped us with the improvement of our manuscript, but suggested some ideas for future studies.

Below you will find our responses to your comments:

  1.  The article utilizes RNA interference (RNAi) to explore the function of the MFE gene, demonstrating its role in ovarian development. However, the control group design might not sufficiently exclude the impact of other potential factors. To enhance the rigor of the study, it is recommended to include additional control groups, such as using non-specific RNA sequences for negative controls and conducting identical experiments across different developmental stages or environmental conditions. This will help verify the repeatability and specificity of the results.

 Response: Thanks for your good suggestion. In this study, we used The GFP (green fluorescent protein) gene in the control group. The prawns in control group of this study were injected with ds-RNA of GFP under the same dose with experimental group. GFP is a specific gene in jellyfish and is widely used as a reporter gene in molecular biology and cell biology researches (Zhang et al. 2016; Gava et al. 2017). 

We add the followed prospect to address your different developmental stages or environmental conditions suggestions.

In our previous study, we performed comparative transcriptomic analysis of the hepatopancreas at 5 ovarian stages and identified many differentially expressed genes related to ovarian maturation. The MFE gene is one of these genes. The main purpose of this study was to investigate the role of this gene in ovarian maturation of M. nipponense. However, this was a preliminary study conducted under experimental conditions. In practical aquaculture applications, environmental factors such as temperature, water quality, diet, and different developmental stages could significantly influence ovary development. The complex polygenic nature of traits can involve genes and environmental interactions. In future studies, we will include environmental factors and different development stages in functional analyses of Mn-MFE to obtain a better understanding of the role of this gene in the rapid sexual maturity of M. nipponense

Zhang, S.B.; Jiang, P.; Wang, Z.Q.; Long, S.R.; Liu, R.D.; Zhang, X.; Yang, W.; Ren, H.J.; Cui, J. DsRNA-mediated silencing of Nudix hydrolase in Trichinella spiralis inhibits the larval invasion and survival in mice. Exp. Parasitol. 2016, 162, 35-42.

Gava, S.G.; Tavares, N.C.; de Matos Salim, A.C.; Gomes de Araujo, F.M.; Oliveira, G.; Mourao, M.M. Schistosoma mansoni: Off-target analyses using nonspecific double-stranded RNAs as control for RNAi experiments in schistosomula. Exp. Parasitol. 2017, 177, 98-103.

  1. The manuscript mentions the use of qPCR and other biological assays, but the description of the statistical analysis methods is relatively brief in the methodology section. To improve transparency and verifiability, it is advisable to provide detailed information about the statistical tests used, significance levels (e.g., P<0.05), and the specific versions and settings of the data processing software. This not only aids in understanding the robustness of the results but also facilitates peer review and future replication studies.

Response: Thank you very much for your suggestion. The statistical tests used, the level of significance, and the specific version and settings of the data processing software have been described in detail in the revised manuscript. Details are as follows: “We calculated the relative expression of genes using the 2–ΔΔCt method. All quantitative data conformed to the homogeneity of variance and the normal distribution, and all data were presented as mean ± standard deviation. The data were then statistically analyzed by one-way analysis of variance and two-tailed t-tests using SPSS 20.0 software (Kang et al. 2018). P < 0.05 indicated significant difference, and P < 0.01 indicated highly significant difference.”

Kang, J.H.; Koo, S.M. Verification of Cervical Cancer Prevention Effect Using SPSS 20.0 WIN Program. Adv. Sci. Lett. 2018, 24, 2166-2170.

Reviewer 4 Report

Comments and Suggestions for Authors

This manuscript examine the effect of the hormone Methyl farnesoate epoxidase on the ovary development of the Macrobrachium shrimp. The result is straight forward but the introduction has not enough background information which make readers feel confused:

1) The authors have not mention the background of this Methyl farnesoate epoxidase hormone. Is this hormone only found in male or female? There are lots of effect of this hormone on crustaceans, for examples, this hormones can affect the sex ratio of the offpsring in Daphinia. See the link below 

https://pubmed.ncbi.nlm.nih.gov/12410602/ 

So, it need a more detail explanation on the nature and function of this hormones in crustaceans and other invertebrates. I think the introduction needed to be extensively rewritten.

2) In the discussion, I think if this hormone is not only specific on female developments, may be the future direction can be use to examine its effect on the sexual dimorphism of males as there are always examples of this genus that the sexual dimorphism can affect the social dominance of the species. Please cite this reference and include the future studies direction of this species:

Author Response

Dear Reviewer 4,

First of all, we are very grateful for your recognition of our research work. Thank you very much for your comments and suggestions. The valuable comments from you not only helped us with the improvement of our manuscript, but suggested some ideas for future studies.

Below you will find our responses to your comments:

  1.  The authors have not mention the background of this Methyl farnesoate epoxidase hormone. Is this hormone only found in male or female? There are lots of effect of this hormone on crustaceans, for examples, this hormones can affect the sex ratio of the offpsring in Daphinia. See the link below

https://pubmed.ncbi.nlm.nih.gov/12410602/

So, it need a more detail explanation on the nature and function of this hormones in crustaceans and other invertebrates. I think the introduction needed to be extensively rewritten.

 Response: Methyl farnesoate is present in both male and female.

Thanks for your good suggestion. We have added more detail to the background of methyl farnesoate in the revised manuscript as follows: “MF can affect the sex ratio of Daphnia offspring by programing embryos to develop into males (Olmstead and Leblanc 2002).MF induced lipid accumulation in hepatopancreas of Scylla paramamosain to prepare for ovarian development (Fu et al. 2022). MF is effective in inducing molting of freshwater crab (Raghavan and Ayanath 2019). Orally administered MF could boost the production of the mature ovary in Scylla olivacea (Muhd-Farouk et al. 2019).” 

Olmstead, A.W.; Leblanc, G.A. Juvenoid hormone methyl farnesoate is a sex determinant in the crustacean Daphnia magna. J. Exp. Zoo. 2002, 293, 736-739.

Fu, Y.; Zhang, F.Y.; Ma, C.Y. Wang, W. Liu, Z.Q.; Chen, W. Zhao, M.; Ma, L.B. Comparative metabolomics and lipidomics of four juvenoids application to Scylla paramamosain hepatopancreas: implications of lipid metabolism during ovarian maturation. Front. Endocrinol. 2022, 13, 886351.

Raghavan, S.D.A.; Ayanath, A. Effect of 20-OH ecdysone and methyl farnesoate on moulting in the freshwater crab Travancoriana schirnerae. Invertebr. Reprod. Dev. 2019, 63, 309-318.

Muhd-Farouk, H.; Nurul, H.A.; Abol-Munafi, A.B.; Mardhiyyah, M.P.; Hasyima-lsmail, N.; Manan, H.; Fatihah, S.N.; Amin-Safwan, A.; Lkhwanuddin, M. Development of ovarian maturations in orange mud crab, Scylla olivacea (Herbst, 1796) through induction of eyestalk ablation and methyl farnesoate. Arab J. Basic App. Sci. 201926, 171–181.

  1. In the discussion, I think if this hormone is not only specific on female developments, may be the future direction can be use to examine its effect on the sexual dimorphism of males as there are always examples of this genus that the sexual dimorphism can affect the social dominance of the species. Please cite this reference and include the future studies direction of this species:

Response: Thanks for your good advice. In our previous study, we performed comparative transcriptomic analysis of the hepatopancreas at 5 ovarian stages and identified many differentially expressed genes related to ovarian maturation. Methyl farnesoate epoxidase gene is one of these genes.  MFE can epoxidize MF to JH and that MF is multi-functional in crustaceans, playing roles in sex differentiation (Olmstead and Leblanc 2002), lipid accumulation (Fu et al. 2022), molting (Raghavan and Ayanath 2019) and promoting ovarian maturation (Muhd-Farouk et al. 2019). In this study, we used RNAi to investigate the role of Mn-MFE only in female ovary maturation. In future steps, we will explore the role of Mn-MFE in males and the results will shed light on the distinct roles of this gene in sexual dimorphism, which can affect the social dominance of the species (Santos et al. 2022).

Olmstead, A.W.; Leblanc, G.A. Juvenoid hormone methyl farnesoate is a sex determinant in the crustacean Daphnia magna. J. Exp. Zoo. 2002, 293, 736-739.

Fu, Y.; Zhang, F.Y.; Ma, C.Y. Wang, W. Liu, Z.Q.; Chen, W. Zhao, M.; Ma, L.B. Comparative metabolomics and lipidomics of four juvenoids application to Scylla paramamosain hepatopancreas: implications of lipid metabolism during ovarian maturation. Front. Endocrinol. 2022, 13, 886351.

Raghavan, S.D.A.; Ayanath, A. Effect of 20-OH ecdysone and methyl farnesoate on moulting in the freshwater crab Travancoriana schirnerae. Invertebr. Reprod. Dev. 2019, 63, 309-318.

Muhd-Farouk, H.; Nurul, H.A.; Abol-Munafi, A.B.; Mardhiyyah, M.P.; Hasyima-lsmail, N.; Manan, H.; Fatihah, S.N.; Amin-Safwan, A.; Lkhwanuddin, M. Development of ovarian maturations in orange mud crab, Scylla olivacea (Herbst, 1796) through induction of eyestalk ablation and methyl farnesoate. Arab J. Basic App. Sci. 201926, 171–181.

Santos, R.C.; Nogueira, C.S.; Jaconis, M.S.; Davanso, T.M.; Costa, R.C.; Hirose, G.L. New insights into the male morphotypes of the amphidromous shrimp Macrobrachium olfersii (Weigmann, 1836)(Caridea: Palaemonidae) and a discussion on social dominance hierarchies. Zoo. Stud. 2022, 61, 83.
